PERSPECTIVE

# Organized Violence and Organized Abandonment Beyond the Human: the Case of Brucellosis among Palestinians in Israel

Osama Tanous,[a,b] Rabea Eghbariah[c]

[a]Department of Public Health Sciences, School of Medicine, University of California, Davis, California, USA
[b]FXB Center for Health and Human Rights, Harvard University, Cambridge, Massachusetts, USA
[c]Harvard Law School, Harvard University, Cambridge, Massachusetts, USA

**ABSTRACT** This article explores how brucellosis became a racialized disease in Israel, where almost all patients are Palestinians. Informed by legal and historical research, the article demonstrates how colonial and settler-colonial policies have targeted Palestinians and their goats and contributed to the distribution of brucellosis along ethno-national lines. Goats, once ubiquitous to the landscape, became enemies of the Israeli state and were blamed for the "destruction" of nature. Under Israeli rule, legal policies not only seized and confiscated Palestinian land but also targeted goat grazing and led to a steep reduction in the number of goats. The resulting depeasantization and concentration of Palestinians in dense poor townships shaped goat grazing as a backyard practice with lack of trust in the hostile state and its brucellosis eradication campaigns. We argue that state policies of organized violence and organized abandonment have shaped the current ecology of brucellosis as a racialized disease.

**IMPORTANCE** The importance of this article is the novelty in combining public health, colonial studies, and legal research to understand the ecology of human brucellosis. This approach allows us to move from a "snap-shot" reading of diseases and cultural practices toward a reading of bacteria, animals, and humans within their political and historical context. The article uses a settler colonial lens to examine the racialized distribution of human brucellosis in Israel and traces colonial policies toward Palestinians and goats—both seen as unwanted intruders to the newly established Israeli nation state. We place these policies in a context of organized violence and organized abandonment, building on the work of Ruth Wilson Gilmore to read the power hierarchies of humans, animals, and diseases and how they shape practices and disease.

**KEYWORDS** brucellosis, Israel, Palestine, colonialism, violence

Human brucellosis (HB) is the most common zoonotic disease globally with over half a million cases annually (1). The disease is transmitted through contact with infected animals or their products (2). Human brucellosis can result in debilitating, long-lasting morbidities and requires prolonged antibiotic therapy. The disease is also a burden to the economy due to its effects on humans, livestock, and wildlife. Despite the great progress in recent decades in the control of the disease due to improvement in sanitation and massive eradication campaigns, the disease is still prevalent in many countries in the global south, as well as in marginalized regions and communities in the global north (1, 3–5).

Brucellosis is ubiquitous in the Mediterranean region and dates back to the time of Hippocrates (6). In Israel, since 1985, all cases of HB result from *Brucella melitensis*, transmitted via goats or sheep (7, 8). This came after the eradication of *Brucella abortus*, transmitted by cows, as a result of a massive S19 vaccination campaign carried out by the veterinary service (9). The incidence of HB varies sharply from 0.2/100,000 in the Jewish population to

Address correspondence to Osama Tanous, otanous@hsph.harvard.edu.

The authors declare no conflict of interest.

33.5/100,000 in the Palestinian population. The Palestinian citizens in Israel (PCIs) are the native Palestinians that remained in their homeland after the majority of Palestinians were expelled in the 1948 Nakba (catastrophe) and were not allowed to return. They became a minority in their own homeland that came to be known as Israel. Currently, the PCI constitute almost 20% of the population, live in mostly segregated towns, and account for 95 to 100% of HB cases (8–10). Among the Palestinian-Bedouin community, a traditionally seminomadic population living in the Naqab desert, the incidence of HB is significantly higher and in 2012 reached a peak incidence of 150/100,000 annually, one of the highest incidence rates globally (11). In two outbreaks of the disease in 2014, only 1 of the 105 patients was Jewish Israeli. The outbreaks occur in areas with large Palestinian communities such as the Galilee, the Naqab desert, and Jerusalem. The only Jewish Israeli patient was reported to have bought unpasteurized cheese from a Bedouin, while some of the Palestinian patients reported buying cheese from "door to door" sales of cheese from family-owned unvaccinated herds (12). HB constitutes a significant health hazard for the PCIs. In a study of 114 hospitalized patients with HB bacteremia, complications were present in 19%, including arthritis, osteomyelitis, and infective endocarditis, and 4 of the 11 pregnant women suffered from spontaneous abortions (13). Ghanem-Zoubi et al. have demonstrated in a cross-sectional study in Israel that localities with a high incidence of HB had significantly higher rates of intrauterine fetal death and preterm birth, among other pregnancy complications, compared to towns with a low incidence (14).

Scholars have attributed the relatively widespread HB among Palestinians to cultural practices (15, 16). In the essay "Writing against Culture," anthropologist Lila Abu-Lughod argues that culture can be a dangerous, populist, and subtle tool used to blame a particular group for a certain type of collective behavior. Cultural explanations rely on generalization to enforce separation, otherness, and hierarchy and thus makes it easier to conceive of a group of people as a discrete, bounded entity, "who do this or that and believe such-and-such" (17). This article will challenge this narrative with regard to HB in Israel and argue that those cultural practices and the resulting abundance of HB among Palestinians is rather exacerbated by the Israeli state's policies toward Palestinians and their livestock. Looking beyond "culture" brings into clearer view how brucellosis has become a racialized disease. (Since race is widely understood as a social construct and not a biological category, we use the term "racialization" here, as explained by Ronit Lentin: "[r]acialization is a technology of the state. It operates by producing a series of distinctions relating to origin, kinship, and lineage, as well as by linking physical characteristics to cognitive abilities, cultural norms, and modes of behavior. Its objective is to propel processes of differentiation and hierarchization in order to facilitate modes of governance and control" [18].) Namely, we argue that HB among Palestinians is best understood through the prism of the state's "organized violence" and "organized abandonment," to borrow Ruth Wilson Gilmore's terms, rather than that of Palestinian cultural practices.

Ruth Wilson Gilmore is a professor of geography who focuses on the study of racism, state power, prisons, and the police. Her analysis of organized abandonment and organized violence explores how states simultaneously use these two tools to further dispossess and control the already impoverished and marginalized. Her work tracks how the state disregards its obligations toward certain people, households, and communities in what Gilmore calls "the anti-state state" and provides unequal levels of support and protection. These same communities that are subjected to organized abandonment are criminalized and marked as undeserving and ineligible for social programs. They are starved of goods and services and subjected to organized violence such as policing, criminalization, and incarceration. This violence causes premature death and produces political power in a vicious cycle. This vicious cycle facilitates an upward mobility of wealth, income, and political power from the relatively poor and powerless to the already rich and powerful. The relatively poor and powerless continue to resist such policies and redefine life in their context (19–21).

We extend this framework of organized violence and organized abandonment, originally used to study police violence and mass incarceration of Black and other communities of color in the United States, not only to humans but also to animals in Palestine/Israel as they became racialized and criminalized in the logic of the settler state. Applying these

terms to the case of HB among Palestinians, we elaborate how the state's organized violence (dispossession, confiscation of lands, home demolition) and organized abandonment (deprivation of social services such as water and electricity infrastructures, education, and health care) (19) provide a more accurate explanation for the proliferation of the disease than merely laying the blame on Palestinian cultural practices.

The Microbes and Social Equity (MSE) Working Group, founded and led by Suzanne Ishaq, Assistant Professor of Animal and Veterinary Sciences at the University of Maine, posits that "microbial exposures across ecosystems, urban and rural settings, and individuals are sociopolitical ... microbiomes have sociopolitical contexts that must be considered" (22). We argue that in order to understand the ecology of HB in Israel, we need to untangle the complex interplay between the pathogen, host, and environment in a settler colonial setting. This article will go further in explaining the inherent mistrust and complex nature of the relationship between the state of Israel, including the veterinary services, and the Palestinian goat owners. Tracking the historical and political forces that shaped this relationship, the article offers a more nuanced and historically responsible explanation for the racialized nature of the disease. This involves paying attention to the history of goat persecution in Palestine/Israel as both animal hosts of the disease and a medium that reflects the logic of settler colonization. We examine this against the background of another Mediterranean location, Malta, where the disease was first discovered. The history of brucellosis eradication in Malta is a fascinating one and is an eye opener for our understanding of the interplay between animals, humans, and colonial armies, where British colonial attempts to change the behaviors and practices of the local Maltese population around goat grazing led to resistance and mistrust. This example shall offer us a prelude to understanding the disease in the current colonial context of Palestine-Israel.

**Local diseases and colonial anxieties: Maltese fever and British soldiers.** Brucellosis was initially called Maltese fever. The disease was devastating to the British Army in the little island causing the hospitalization of thousands of soldiers for long periods and inflicting a heavy economic burden (23). The British Royal Army Medical Corps and the Mediterranean Fever Commission led a series of pioneering research discoveries that identified *B. melitensis* as a pathogen and the goat as a vector transmitting the disease through milk and cheese. This has led to a series of regulations with the banning of raw goat milk consumption. The disease was completely eliminated among the British soldiers in 1908, as they were forbidden from consuming goat milk and shifted to tinned milk (24). As for the local Maltese population, who have been accustomed to drinking fresh goat milk, the matter was much more complicated. The prevalence of HB continued to increase among the local Maltese population. Attempts by the British to ban the selling of goat milk to cafes and hotel were unsuccessful (25). People were not compliant with regulations to slaughter the infected goats. The Maltese had negative attitudes toward the imposition of regulation that aimed to change their taste and their relationship with their beloved goats (26). Regulation on licensing and hygiene rules were met with resistance, strikes, and noncompliance. The local Maltese population saw the regulations and legislations as compulsive measures from outsiders who do not understand or care about the hardships and life challenges of the impoverished goat owners and do not understand the generational ties to goat milk and goat herding (23). Small herd owners found ways to bypass regulations and would smuggle goats to neighbors before inspections, while goat milk consumers found ways to purchase milk directly from vendors when goats were forbidden to enter cities and villages. Poverty and lack of development and sanitation were major obstacle toward proper hygiene of the farms. The epidemic for the local Maltese ended only in the 1990s after independence, economic development, and new regulations and massive education campaigns reaching the entire population (25). Tripp and Sawchuk argue that "a complex interplay of colonialism acting on cultural milk practices together with the scale effect's impact on the creation, implementation, and enforcement of health policy" can help us explain the challenges in eradicating HB in Malta (23).

We argue that in Palestine/Israel, the outbreaks of HB among Palestinians cannot be fully understood without considering the Israeli practices toward goats and goat

herding and the systematic impoverishment and dedevelopment of the Palestinian economy and livelihood, as well as the structural conditions pertaining to sanitation and hygiene. (Note that we use the term "dedevelopment" as articulated by Sarah Roy as a structural relationship where a dominant economy not only distorts the development of the subordinate economy but undermines it entirely [27]) In order to untangle this complex interplay, we will draw on historical materials to explain British colonial and later on Israeli attitudes toward goats in Palestine and then explore Israeli state violence toward Palestinian Bedouins in the Naqab. We will conclude by showcasing how these factors have shaped the racialized distribution of HB.

**Settler colonialism, beyond the human.** Unlike colonialism, settler colonialism seeks not only to dominate the colonized population and natural resources but also to replace the native population with a new population of settlers. It is an ongoing process rather than an event, and it operates within the "logic of elimination," to use Patrick Wolfe's terms (28). This process is not limited to the alteration of human demography: European settler colonialism aimed at creating "new Europes" in the settler colonies (North America, Australia, and New Zealand) through transforming the landscape, animals, and flora. These colonial interventions proved to be not only catastrophic for the indigenous communities but also for the environment and ecosystem (29, 30).

Similarly, the Israeli settler colonial project also was not limited to altering the human demography of the land. In addition to creating a Jewish majority at the expense of Palestinians by means of ethnic cleansing and mass expulsion (31), Zionist and Israeli policies aimed to redefine the natural landscape, waterscape (32), and even the livestock composition. Palestine was often portrayed by the Zionists as a land of wilderness, deserts, and swamps that is underdeveloped and with underutilized absorptive capacity for settlers. It was seen as a land that is dirty, lacking trees, and in need of hygienic redemption and reengineering (33, 34).

The Naqab desert, a place where Palestinian-Bedouins have lived for hundreds of years and made sustenance from a dual economy of farming and grazing, became a target of "Europeanization," "beautification," and "blooming" (35). The first Israeli Prime Minister David Ben-Gurion has written, for example, "the Negev (Naqab) is a desolate area which is currently empty of people. . .. it lacks water and Jews. . . two million Jews can be settled there with agriculture and two millions with industries" (inspired by "Ben-Gurion"). In 1948, the year marking the Palestinian Nakba, the majority of Bedouins were expelled from their homes and homeland as part of the broader ethnic cleansing of Palestine (36). Only 13,000 of 95,000 Palestinian-Bedouins remained in the Naqab area and were confined to a closed military zone known as "al-siyaj" (37). Since then, Bedouins have been subjected to violent state policies of land confiscations, home demolitions, and dedevelopment. Today, approximately half of the Bedouin population in the Naqab live in so-called "unrecognized villages," namely, localities that are unrecognized by the Israeli state and therefore lack basic infrastructures such as health care, water, and electricity. Others live in planned townships, organized by the state in an attempt to concentrate as many Bedouins on as little land as possible. These townships became microcosms of poverty, unemployment, and crime (38, 39).

In the new reality, not only the Bedouins were not welcomed and became "trespassers" on state lands, but also their goats rapidly became the target of Israeli state policies. Goats were the most abundant livestock in Palestine before Zionist colonization materialized in the form of Israel in 1948. While only 3% of Palestinians were exclusively engaged in animal husbandry as their sole source of income by that time, goats and their dairy products were an important source of subsistence, especially in the rural economy. In the early 1940s, for example, the goat population reached a peak number of some 625,000, owned predominantly by Palestinians (40).

The increase in goat numbers during the British Mandate years, from 1917 to 1948, however, was not welcomed by the British administration. Influenced by orientalist imaginations and perceptions of the local landscape, the British colonial administration sought to control, engineer, and replace what they considered desolate and underdeveloped landscape. Afforestation projects (i.e., the planting of trees to establish human-made forests)

proved to be a powerful tool to seize control over the land and population under the pretext of development. However, the goats found these newly planted trees palatable and soon became the forests' number one enemy. In this context, certain ideas about the local goats grew particularly popular among the British administration, who accused the goats of causing soil erosion, flooding, and desertification. Officials who disputed these claims from within the administration were ignored and dismissed. Since the 1930s, the British administration enacted several policies to reduce the goat population and disincentivize its ownership, with little to no actual success (40, 41).

European Zionists who settled in Palestine during the British Mandate years harbored similar sentiments toward the landscape and the goats. After seizing control through force and establishing the state of Israel, Israeli afforestation projects intensified, and the Israeli legislature soon moved to target the goats. In 1950, the Knesset discussed during several sessions the goat "problem" and sought to achieve the "termination of goat herds." These discussions eventually culminated in the Plant Protection Law (Damages of Goats), also known as the Black Goat Act. The proposed bill stated that "the existence of many goat flocks is a major obstacle to the forestation projects taking place in large-scales these days. The proposed law limits the possibility of possessing goats." Article 2 stressed that "a human shall not possess or graze goats except on lands in his tenure and according to the allowed ratio," and article 1 defined the "allowed ratio" as "one goat for each 40 dunams." The law additionally provided state officials the authority to seize goats and criminally prosecute their possession and grazing. In reality, however, the practice of goat rearing continued, and the Israeli state faced many hurdles in implementing the law in the 1950s and 1960s. During this period, the law placed the goat and its Palestinian owners in liminal legality: while criminal prosecution and herd confiscations did not go into full force, the goats were left abandoned to graze in the field of supposed illegality with no state attention or resources directed at preventive health measures. While the law did not succeed immediately in reducing the goat population owned by Palestinians, the massive confiscation of Palestinian-owned lands during the 1950s and the subsequent concentration of 93% of the land in state property, coupled with the military rule imposed on Palestinian citizens of Israel between 1948 and 1966, had insurmountable ramifications on Palestinians' ability to maintain pasture and agricultural activities. Losing property and access to land pushed the Palestinian fellahin (peasants) and goat-owners from the field into the city and repositioned them as a cheap labor force in the Israeli economy. This imposed loss of the means of production—land and livestock—amounted to an increasing proletarianization and depeasantization of Palestinian citizens of Israel (40–42).

Those who nonetheless managed to maintain goats, especially some Bedouins communities in the Naqab area, were later subjected to sanctions for illegal goat grazing or possession. In 1976, the Israeli government established a unit called the "Green Patrol" and revived the plans to terminate the goats under the force of the already existing law. Starting from the late 1970s, the practice of goat grazing became a risky endeavor and exposed the goat owners to clashes with Israeli state authorities who relentlessly chased goats and their owners. Goat numbers in "Israel proper" decreased sharply from over 100,000 goats in the 1970s to less than 20,000 goats by the 1980s. Since then, the number of goats has not grown any further. Despite these harsh measures, Palestinian Bedouins continued to graze goats in the shadow of the law and in an act that could be best understood as an "everyday form of resistance" (to borrow a term from James Scott) (41, 43).

The implementation of the law turned goat rearing and pasturing into a backyard practice with even less accessibility to preventive health measures. Even when the persecution of goats decreased at the end of the century, goats and their owners were still not prioritized in public policies and had unfavorable accessibility to allocated grazing land or adequate veterinary supervision. The Black Goat Act, which criminalized goat grazing and possession, was officially repealed by the Israeli legislature only in 2018 after several massive scale fires swept the forests planted by the Jewish National Fund. Scientific evidence supported the role of the goats in reducing the risk of fires by eating flammable bushes and shrubs. Jamal Zahalka, a

Palestinian parliamentary member who pushed the repeal of the law, stated that this will "restore the goats lost honor" and "repair a historic injustice" for Palestinian farmers (44).

**Hierarchies of humans, animals, and dairies.** In order to fully understand the racialization of HB, and generally the unequal distribution of diseases, we argue that it is of utmost important to examine the racialization and hierarchies of humans, animals, and in our case the dairy industry. As in every settler colonial process, the settler views himself as "racially" superior and thus claims sovereignty over the indigenous land and resources and aims to transform the landscape. In the colonization of Palestine, the Jewish settlers during the British Mandate, later on, Jewish Israelis view themselves as the sovereign nation living in their natural nation state. With time, the settler becomes to view himself as native (45). The indigenous Palestinian, on the other hand, is seen as an intruder. A tolerated guest at best and an intruding enemy or a security threat to be punished. Also, when Palestinians that remained in Israel were granted citizenship, that citizenship was used as a form of colonial domination. The settlers are seen as natural and authentic citizens, while the indigenous Palestinians are seen as a naturalized citizens, aliens and guests in their homeland (46).

The same hierarchy followed the livestock. Cows were historically rare in Palestine due to the limited water resources and hot climate. The Zionist ethos of a land of "milk and honey" has led to the centralizing of cow-based, industrial dairy production in the settler economy. The Israeli Holstein cow was created as a breed between the local Damascus cow and the Dutch Holstein cow to be able to produce large amounts of milk and tolerate the hot and humid weather. This dairy industry became one of the cornerstones of the Israeli economy as it developed to be fully regulated, industrialized, and vaccinated (42, 47). *Brucella abortus*, transmitted by beef and dairy cattle, was eradicated in Israel in 1984 following a successful test and slaughter campaign with proper compensation to farmers and a vaccination campaign (9, 48). The Israeli Holstein cow became the dominant homogenous milk producer animal.

The goat on the other hand, was seen as an inferior animal, an intruder and destructive to nature and the environment. It was declared an official enemy of the state and was put early on the list for eradication. The dairy production from goats was not developed to an industrial production. Palestinians had no choice but to continue the practice of goat grazing as a backyard practice of small herds and small flocks, practiced in the shadow and hidden from the eyes of the state and the veterinary services. The Palestinians also were pushed to the bottom of the socioeconomic ladder. The ones that remained in their homeland and became Israeli citizens were put under military rule, had their lands confiscated, and were transformed from farmers to landless peasants without the ability to make a living from agriculture or herd grazing. Until the present day, most Palestinians in Israel live in segregated towns that are at the bottom of the socioeconomic grading. These towns became areas of segregated and racialized poverty and unemployment (49, 50). Palestinian workers are often concentrated in low-paying sectors of the economy such as construction and industries (51) that carry high occupational health hazards such as falls, injuries, and death (52).

In the Naqab, Where HB is most endemic, more than 230,000 Bedouin live, the vast majority in townships of racialized poverty. Half of the villages are not recognized by the state and lack basic infrastructures of electricity and running water. They are subjected to a regime of organized abandonment when it comes to infrastructures, health and welfare, and direct state organized violence that manifest in home demolition, land confiscation, and policing (37–39). These factors contribute to the lack of hygiene and sanitation in farming and make the organizing of dairy production and refrigeration impossible.

**A way forward.** The "One Health" approach urges us to develop a holistic understanding toward the health of humans, animals, and ecosystems and the complex interplay between them. It pushes us to understand how political and economic processes such as industrialization and political instability influence the health of humans, animals, and the entire environment and the need for a policy framework that encompasses public health, farming, and agriculture (53). Our study further expands the understanding of alteration in biodiversity, ecosystem dynamics, and habitat destruction, often used in a "One Health" approach, not as mere byproducts of capitalism, industrialization, and domestication of animals, but as direct and deliberate targeting by settler colonial logic and violence aiming to alter the

environment, its animals, and humans to fit a colonial ethos. This targeting reshapes the landscape and distribution of zoonotic and human diseases, having some treated efficiently and other neglected and thus creating racialized humans and diseases. Our research elucidates the importance of incorporating political and historical processes like colonialism and settler colonialism when researching zoonotic diseases in the global south and settler colonial states. These processes not only affect colonized humans but immensely reshape and distort ecosystems, including animals, landscapes, and waterscapes, in the areas they dominate, turning nature into "natural resources" to generate commodities and profit in global capital networks (54, 55).

Studies conducted in the global south have demonstrated the promising potential of community-based brucellosis control program that include multiple stakeholders and center education, in altering human risky behaviors (56). Furthermore, a proper implementation of the complex approach of "One Health," combined with tools of "theory of change" and "outcome mapping" have demonstrated the ability to guide and monitor complex interventions. The successful example of "SafePORK" in Vietnam demonstrates the immense potential of using theory to guide the mapping of food production, distribution, and consumption chains and identifying targets for interventions aiming to improve food safety and prevent zoonotic infections (57).

In Israel, the veterinary services oversee HB control. It is carried through Rev1 vaccination of young females and annual serological testing of unvaccinated males. When HB is reported, the herd is extensively tested and sick animals are eradicated (8). A national intervention program for the control of livestock brucellosis was initiated in 1994 with some initial success; 40,000 animals were slaughtered, and more than 72% were declared free of disease. Despite this partial success, the program was terminated in 1997 due to declared lack of budgets and failure to eradicate the disease. Between 2002 and 2015, the HB reemerged and continued to increase in prevalence among the Bedouins and later among all PCI. Ghanem et al. have demonstrated that the disease was limited to the Naqab between 2010 and 2013 and later on spread to other Palestinian localities across the country. Her work emphasized the missed opportunity to control the disease and the unsatisfactory function of the veterinary service in Palestinian localities (9, 11). Jaber et al. have reported a promising experience in the city of Taibe, a large Palestinian city in Israel, where a low-budget, multidisciplinary, community-based program involving local physicians, veterinarians, nurses, school officials, and health inspectors in 1994 have conducted intensive public health education campaign and periodic examination and vaccination of animals. The campaign has resulted in a sharp decline in the incidence of the disease, from 175.0/100,000 in 1993 to 5.7, 10.4, and 2.5/100,000 in 1994, 1995, and 1996, respectively (odds ratio = 24.44; $P < 0.0001$). The campaign has also resulted in an impressive increase in residents' awareness of brucellosis and preventive measures. They have concluded that HB is a preventable disease and that an intersectoral community-based collaboration is an important tool for controlling the disease (58). The successful examples from Taibe, as well as Malta and other settings, prove that the eradication of HB is possible and doable, even with budget constraints. We believe it is feasible and necessary to conduct a brucellosis eradication program in Israel. In order to do this, it is crucial to recognize and treat the root causes of the disease and its racialized distribution. It is crucial to break the cycle of mistrust between goat owners and the authorities and have state actors recognize the harms they have committed and work toward a holistic intervention program of improving testing, vaccination, and infrastructure building for small-scale goat farms that produce goat milk, yogurt, and cheese, all conducted in an environment of trust without fear of punishment. In the meantime, both the Palestinians and goats continue to lie in a toxic geography shaped by state power and indifference (21). The Palestinian and the goat have been, and continue to be, subjected to organized violence and organized abandonment (19) as a part of the larger structural violence and structural racism that shapes the life of Palestinians in Israel, similarly to other indigenous peoples in other settler colonial settings.

**On borders and diseases.** Humans, animals, and zoonotic diseases are not static and cross state borders. Unlike Europe, where unauthorized trade in animals and their products occurs across sovereign state borders and the suggested solutions are ones

of state cooperation (59), in our case, such trade and transfer of goats and their food product occurs across the borders of colonizer and colonized: Israel and the occupied West Bank (7). In such a reality, where the West Bank is militarily occupied and economically captive (60), a decent cross-border cooperation built on trust is an unrealistic scenario. What does happen on the ground is a unilateral export of goats and sheep from Israel to the occupied West Bank and a parallel unauthorized and undocumented bidirectional smuggling of animals and agricultural products between Bedouin communities estimated at hundreds of tons of meat annually (7). While these borders are highly militarized and surveilled (61), we believe that colonial indifference allows such large-scale smuggling to occur since it does not damage it directly and mainly damages colonized Palestinians on both sides of the borders. Whereas a justice-based approach and sound logic would call for decolonization, the end of military occupation and economic subjugation and then cooperation, what is being discussed is the construction of even more hermetic borders, as can be understood from a governmental veterinarian quoted by Hermesh et al. "Smuggling does happen, and I expect that the whole southern Hebron Mountain will be closed with a fence" (7).

Progressive scholars are leading discussions on decolonizing health care and reparation as tool to achieving health equity (62, 63). If we want to move forward with the eradication of HB among Palestinians, it is crucial to move beyond blaming marginalized and colonized communities for their behaviors or cultures, in itself an act of symbolic violence, and address the historical injustices toward humans and goats. Only then we can think of creating justice-based, community-centered initiatives and eradicate the disease.

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
