## [Reviewer comments · mSystems]

Organized Violence and Organized Abandonment Beyond the Human; The Case of Brucellosis among Palestinians in Israel

osama Tanous and Rabea Eghbariah

Corresponding Author(s): osama Tanous, University of California, Davis

Review Timeline:

Submission Date:	January 3, 2022
Editorial Decision:	January 15, 2022
Revision Received:	February 4, 2022
Editorial Decision:	February 21, 2022
Revision Received:	March 13, 2022
Accepted:	March 16, 2022

Editor: Suzanne Ishaq

Reviewer(s): Disclosure of reviewer identity is with reference to reviewer comments included in decision letter(s). The following individuals involved in review of your submission have agreed to reveal their identity: Trude Bennett (Reviewer #1); Min Yue (Reviewer #3)

Transaction Report:

DOI: <https://doi.org/10.1128/msystems.01499-21>

January 15, 2022

Dr. osama Tanous
University of California, Davis
School of Public Health
One Shields Avenue.
Medical Sciences Bldg. 1C
Davis, CA 95616

Re: mSystems01499-21 (Organized Violence and Organized Abandonment Beyond the Human; Hierarchies of Humans, Animals and Diseases - The Case of Brucellosis among Palestinians in Israel)

Dear Dr. osama Tanous:

Thank you for submitting your manuscript to mSystems. We have completed our review and I am pleased to inform you that, in principle, we expect to accept it for publication in mSystems. However, acceptance will not be final until you have adequately addressed the reviewer comments.

Preparing Revision Guidelines

Sincerely,

Suzanne Ishaq

Editor, mSystems

Journals Department
Reviewer comments:

Reviewer #1 (Comments for the Author):

This is a highly original and fascinating paper that succeeds in situating a harmful bacterial disease, Human Brucellosis (HB), in its full historical, political, legal, socioeconomic, cultural, and behavioral context to explain extreme social inequalities in the disease in a particular setting. Whereas many attempts are made to define the multi-causal origins of disease using a bio/sociocultural model, this paper captures the upstream dynamics that must be considered to achieve disease prevention and reduce inequalities. The findings are significant in themselves, and also as a model for penetrating and interpreting causality in both settler colonial and other situations. The paper is well organized and fully referenced, and the conclusions are sound and compelling.

Given the target audience of the journal (who may not be well schooled in the social sciences), the use of the term "racialization" may be somewhat confusing or misleading. To clarify, in the text or in a footnote, that "race" or "racialization" is not meant to imply biological or genetic characteristics, the authors might want to explain the terminology or substitute a different term such as "marginalization," "discrimination," or "colonization." Similarly, while the power analysis of Ruthie Wilson Gilmore is a very appropriate theoretical framework, it would be helpful to summarize briefly Dr. Gilmore's work.

A specific reference for the estimate of half a million cases in lines 72-73 would be helpful (even if included in Refs 1-4 at the end of the paragraph). Are any estimates available for the prevalence of Brucellosis in goats or other animals without eradication efforts?

The term de-development is somewhat intuitive, but a definition might be useful.

Line 96: Is it likely that HB is under-diagnosed in women having spontaneous abortions in this or similar settings?

Line 103: The situation you describe is not "either/or" -- can you suggest or foreshadow here the interrelationship between state policies and cultural practices?

Line 104: Does the Microbes and Social Equity (MSE) Working Group? have a geographical location or institutional affiliation?

Line 121: The Maltese example is so interesting and instructive, as is the later contrast with the breeding of Israeli Holstein cows.

Line 187: Does this mean "exclusively engaged"?

Line 192: State the years of the British Mandate.

Line 195: Perhaps define "afforestation" (it is clear in context, but probably an unfamiliar term).

Line 198-200: Was there established science at the time to refute these claims?

Line 235: Are more recent (since 1980) estimates available?

Line 237: Perhaps set off in quotes "everyday form of resistance"?

Line 240: Persecution or prosecution?

Line 253: Put "racially" in quotes, or just superior?

Lines 280-282: For those relocated Palestinians who have employment, what are common occupations (and associated health hazards)?

Line 295: Can you clarify -- were the 40,000 slaughtered animals shown to be diseased or was this a "preventive" culling? Did the 40,000 compose 28% of the total, with the other 72% free of disease?

Lines 301-311: Can you say what conditions made possible this successful, community-based collaborative effort in Taibe?

Reviewer #2 (Comments for the Author):

Thank you for submitting your article for publication. This is an important subject and a unique perspective. Provided below are some of the details that should be addressed in the manuscript.

Minor revisions

- there is no need to capitalize the B in brucellosis, except when referring to the scientific name, *Brucella* spp.
- line 32, Palestinian should be plural.
- line 49 appears to be a smaller (or maybe not bolded?) font size compared to neighboring lines.
- line 49, should reding be reading?
- line 52, goats does not need to be capitalized
- line 75, disease does not need to be capitalized
- line 81, *Brucella Melitensis* should be *Brucella melitensis*
- line 82, *Brucella Abortus* should be *Brucella abortus*
- line 82, sheep does not need the "s" to be plural
- line 84, is it 0.2/100 or 0.2/100,000? Same question for 33.5/100 or 33.5/100,000? Line 89 refers to the incidence out of 100,000, so my guess is that it should state 02. and 33.5 out of 100,000, but there is currently a decimal instead of a comma in 100,000.
- line 93, 100, 102 (and many other lines in paper): the font size appears to be smaller or not as bold in some of these sentences. Please check for consistency throughout the paper.
- line 122, it should be brucellosis instead of *Brucella*
- line 125, *B. melitensis* should be italicized.
- line 130, could say "fresh goat milk" instead of freshly milked goat milk
- lines 130-139, are there additional citations/references that can be used throughout this section, or is it all related to reference number 16?
- lines 136-139, please check verb tense and ensure consistency (line 138, doesn't should be don't/do not)
- line 173, could state "the first prime minister of Israel" or "Israel's first prime minister."
- line 185, could state "their goats rapidly became the target. . ." instead of "but also their goats who rapidly became the target. . ."
- line 187, per cent should be percent; depending on the journal specifications, could also be written as 3%.
- lines 192-203, are all details covered by references 32 and 33?
- line 198, goat should be plural
- line 223, should state between 1948 and 1966 (or from 1948 to 1966) instead of "between 1948 to 1966."
- line 228, remove "a" from end of proletarianization
- line 233, delete "put"
- line 233, do not need hyphen in goat-owners
- lines 241-242, check word choice, tense and order
- line 246, reduce should be reducing
- line 247, sentence could read ". . .a Palestinian parliamentary member pushed the repeal. . ."
- line 260, remove "a" after "as"
- line 261, "is" should be "are"
- line 262, could just say "Cows" instead of The Cows
- line 264, could add hyphen to cow based
- line 267, should say "became a cornerstone of. . ." or "one of the cornerstones" and could replace the second use of "became" in sentence with "as it developed into. . ."
- line 268, *Brucella abortus* is the appropriate spelling.
- line 269, replace "has been" with "was"
- line 275, add "s" to Palestinian
- line 275, the authors toggle between "backyard" and "back yard." Select one and be consistent.
- line 277 and 281, authors toggle between socio-economic and socio economic. Could also use socioeconomic. Select one and be consistent.
- line 278, Israel should be Israeli
- line 280-282, please check word choice, tense and order
- line 283, use 283,000 instead of 283 thousand
- line 315, should it be "both the Palestinian and the goat"?

Major revisions

- Please ensure that font size and other formatting is consistent throughout the paper, as there are multiple places throughout the paper where the font size appears to change.
- Please ensure that the spelling of *Brucella* spp. or brucellosis is used appropriately throughout the paper.
- Additional use of citations in certain parts of the paper may be warranted.

Article Summary

This paper supports the theory that the status of human brucellosis cases in Israel is the result of historical racial tensions and persecution of Palestinians by the Israeli government, rather than due to cultural practices. The authors provide a thorough history of the marginalization of the Palestinian Bedouins, and subsequently their goats, and how these communities were forced to raise their herds as an illegal backyard practice rather than openly as a profession. Thus, limiting appropriate preventive veterinary care of the goat herds and increasing public health risk of brucellosis. The authors also cite recent brucellosis case numbers and a previously implemented successful control program. They urge that addressing the colonization of these communities is necessary to address brucellosis at the population level.

Minor revisions

- there is no need to capitalize the B in brucellosis, except when referring to the scientific name, *Brucella* spp.
- line 32, Palestinian should be plural.
- line 49 appears to be a smaller (or maybe not bolded?) font size compared to neighboring lines.
- line 49, should reding be reading?
- line 52, goats does not need to be capitalized
- line 75, disease does not need to be capitalized
- line 81, Brucella Melitensis should be *Brucella melitensis*
- line 82, Brucella Abortus should be *Brucella abortus*
- line 82, sheep does not need the "s" to be plural
- line 84, is it 0.2/100 or 0.2/100,000? Same question for 33.5/100 or 33.5/100,000? Line 89 refers to the incidence out of 100,000, so my guess is that it should state 02. and 33.5 out of 100,000, but there is currently a decimal instead of a comma in 100,000.
- line 93, 100, 102 (and many other lines in paper): the font size appears to be smaller or not as bold in some of these sentences. Please check for consistency throughout the paper.
- line 122, it should be brucellosis instead of Brucella
- line 125, B. melitensis should be italicized.
- line 130, could say "fresh goat milk" instead of freshly milked goat milk
- lines 130-139, are there additional citations/references that can be used throughout this section, or is it all related to reference number 16?
- lines 136-139, please check verb tense and ensure consistency (line 138, doesn't should be don't/do not)
- line 173, could state "the first prime minister of Israel" or "Israel's first prime minister."
- line 185, could state "their goats rapidly became the target. . ." instead of "but also their goats who rapidly became the target. . ."
- line 187, per cent should be percent; depending on the journal specifications, could also be written as 3%.
- lines 192-203, are all details covered by references 32 and 33?
- line 198, goat should be plural
- line 223, should state between 1948 and 1966 (or from 1948 to 1966) instead of "between 1948 to 1966."

- line 228, remove “a” from end of proletarianization
- line 233, delete “put”
- line 233, do not need hyphen in goat-owners
- lines 241-242, check word choice, tense and order
- line 246, reduce should be reducing
- line 247, sentence could read “. . .a Palestinian parliamentary member pushed the repeal. . .”
- line 260, remove “a” after “as”
- line 261, “is” should be “are”
- line 262, could just say “Cows” instead of The Cows
- line 264, could add hyphen to cow based
- line 267, should say “became a cornerstone of. . .” or “one of the cornerstones” and could replace the second use of “became” in sentence with “as it developed into. . .”
- line 268, *Brucella abortus* is the appropriate spelling.
- line 269, replace “has been” with “was”
- line 275, add “s” to Palestinian
- line 275, the authors toggle between “backyard” and “back yard.” Select one and be consistent.
- line 277 and 281, authors toggle between socio-economic and socio economic. Could also use socioeconomic. Select one and be consistent.
- line 278, Israel should be Israeli
- line 280-282, please check word choice, tense and order
- line 283, use 283,000 instead of 283 thousand
- line 315, should it be “both the Palestinian and the goat”?

Major revisions

- Please ensure that font size and other formatting is consistent throughout the paper, as there are multiple places throughout the paper where the font size appears to change.
- Please ensure that the spelling of *Brucella* spp. or brucellosis is used appropriately throughout the paper.

Authors' Responses to Reviewers 1 and 2

Date: 20 January 2022

Manuscript Title: Organized Violence and Organized Abandonment Beyond the Human; Hierarchies of Humans, Livestock, Dairies and Diseases. The case of Brucella among Palestinians in Israel

Revised Title: Organized Violence and Organized Abandonment Beyond the Human; The case of Brucella among Palestinians in Israel

The authors would like to sincerely thank the reviewers for their deep, thorough reading of the manuscript and their constructive feedback and suggestions. These thoughtful comments and suggestions have pushed us to extensively edit and sharpen our manuscript. We are resubmitting a version with highlighted changes.

Kindly see below our point-by-point responses to the reviewers' comments.

Reviewer #1 (Comments for the Author):

1. This is a highly original and fascinating paper that succeeds in situating a harmful bacterial disease, Human Brucellosis (HB), in its full historical, political, legal, socioeconomic, cultural, and behavioral context to explain extreme social inequalities in the disease in a particular setting. Whereas many attempts are made to define the multi-causal origins of disease using a bio/sociocultural model, this paper captures the upstream dynamics that must be considered to achieve disease prevention and reduce inequalities. The findings are significant in themselves, and also as a model for penetrating and interpreting causality in both settler colonial and other situations. The paper is well organized and fully referenced, and the conclusions are sound and compelling.

Response: We thank the reviewer for the kind words.

2. Given the target audience of the journal (who may not be well schooled in the social sciences), the use of the term "racialization" may be somewhat confusing or misleading. To clarify, in the text or in a footnote, that "race" or "racialization" is not meant to imply biological or genetic characteristics, the authors might want to explain the terminology or substitute a different term such as "marginalization," "discrimination," or "colonization." Similarly, while the power analysis of Ruthie Wilson Gilmore is a very appropriate theoretical framework, it would be helpful to summarize briefly Dr. Gilmore's work.

Response: We agree with the reviewer and thank them for this comment. We have added the following footnote to line 110 “As race is being widely understood as a social construct and not a biological category, we will use the frame “racialization” as explained by Ronit Lentin “[r]acialization is a technology of the state. It operates by producing a series of distinctions relating to origin, kinship, and lineage as well as by linking physical characteristics to cognitive

abilities, cultural norms, and modes of behavior. Its objective is to propel processes of differentiation and hierarchization in order to facilitate modes of governance and control.”

(Lentin 2017)

We have added the following summary of Dr. Gilmore’s work in line 114:

Ruth Wilson Gilmore is a professor of geography who focuses on the study of racism, state power, prisons, and the police. Her analysis of organized abandonment and organized violence explores how states simultaneously use these two tools to further dispossess and control the already impoverished and marginalized. Her work tracks how the state divorces its obligations towards certain people, households, and communities in what she calls “the anti-state state” and provides unequal levels of support and protection. These same communities that are subjected to organized abandonment are criminalized and marked as undeserving and ineligible for social programs. They are starved of goods and services and subjected to organized violence such as policing, criminalization and incarceration. This violence causes premature death and produces political power in a vicious cycle. This vicious cycle facilitates an upward mobility of wealth, income and political power from the relatively poor and powerless to the already rich and powerful. The relatively poor and powerless continue to resist such policies and redefine life in their context (16–18).

We extend this framework of organized violence and organized abandonment, originally used to study police violence and mass incarceration of Black and other communities of color in the USA, not only to humans but also to animals in Palestine/Israel as they became racialized and criminalized in the logic of the settler state. Applying these terms to the case of HB among Palestinians, we elaborate how the state’s organized violence (dispossession, confiscation of lands, home demolition) and organized abandonment (deprivation of social services such as

water and electricity infrastructures, education and healthcare) (16) provide a more accurate explanation for the proliferation of the disease than merely laying the blame on Palestinian cultural practices.

A specific reference for the estimate of half a million cases in lines 72-73 would be helpful (even if included in Refs 1-4 at the end of the paragraph). Are any estimates available for the prevalence of Brucellosis in goats or other animals without eradication efforts?

Response: The half million estimate comes from reference #1. We have added that directly after the estimate in line 73. In page 21 in the section on epidemiology of Brucellosis in animals, the WHO report on brucellosis in humans and animals (“Brucellosis in Humans and Animals Food and Agriculture Organization of the United Nations. 2006) states that: “Latent infection has been estimated to occur in the progeny of about 5% of infected cows. The extent of the problem in other species is not known, but latency has been documented in sheep.”. so, we don’t have estimates of the prevalence of Brucellosis in goats and assume it changes dramatically by the availability of screening methods and location.

The term de-development is somewhat intuitive, but a definition might be useful.

Response: We thank the reviewer for this remark. We have added the following footnote in line 183 “We use the term de development as articulated by Prof Sarah Roy as a structural relationship where a dominant economy not only distorts the development of the subordinate economy but undermines it entirely.(Roy 1999)

Line 96: Is it likely that HB is under-diagnosed in women having spontaneous abortions in this or similar settings?

Response: We thank the reviewer for this remark. Yes, indeed this is likely. We have added the following sentence with a reference to a study that focused of pregnancy outcomes of HB in line 97 “ Ghanem Zoubi et al have demonstrated in a cross sectional study in Israel that localities with high incidence of HB had significantly higher rates of intra uterine fetal death and preterm birth among other pregnancy complications compared to towns with low incidence (Ghanem-Zoubi et al. 2018).”

Line 103: The situation you describe is not "either/or" -- can you suggest or foreshadow here the interrelationship between state policies and cultural practices?

Response: We thank the reviewer for this remark. We have further explained how using the term “culture” in order to explain collective behaviors can be a dangerous tool. We have added the following paragraph in line 102-109 In “writing against culture”, the anthropologist Lila Abu-Lughod argues that culture can be a dangerous, populist, and subtle tool used to blame a particular group for a certain type of collective behavior. Cultural explanations rely on generalization to enforce separation, otherness and hierarchy and thus makes it easier to conceive of a group of people as a discrete, bounded entity, “who do this or that and believe such-and-such” (15). This article will challenge this narrative with regards to HB in Israel and argue that those cultural practices and the resulting abundance of HB among Palestinians is rather exacerbated by the Israeli state’s policies towards Palestinians and their livestock. Looking beyond “culture” brings into clearer view how brucellosis has become a racialized disease.

Line 104: Does the Microbes and Social Equity (MSE) Working Group? have a geographical location or institutional affiliation?

Response: We thank the reviewer for this reminder. We have added “founded and lead by Prof Suzanne Ishaq from the University of Maine” in line 135.

Line 121: The Maltese example is so interesting and instructive, as is the later contrast with the breeding of Israeli Holstein cows.

Response: Thank you. Indeed, it is a fascinating and educational example.

Line 187: Does this mean "exclusively engaged"?

Response: Yes. We have changed the wording accordingly. It is now in line 223.

Line 192: State the years of the British Mandate.

Response: We thank the reviewer for this reminder. We have added “from 1917 to 1948” in line 228.

Line 195: Perhaps define "afforestation" (it is clear in context, but probably an unfamiliar term).

Response: we have edited the paragraph and explained the term “afforestation” by adding: (i.e. the planting of trees to establish human-made forests) in line 231.

Line 198-200: Was there established science at the time to refute these claims?

Response: We edited the sentence to avoid describing the evolution of the science behind these claims. Instead, we decided to rephrase the framing to reflect dispute within the administration’s officialdom more broadly, without any explicit reference to the scientific claims made at that point.

[For context: these claims were both articulated and disputed by officials at the British administration during the first half of the 20th century. Some of these officials also acted in their role as “scientific experts” (such as the chief of veterinary services, chief of forest division, etc.). In the 1970s, several Israeli scientists publicly contested the Israeli government’s claims that goats pose a threat to nature, and only in the 1980s and 1990s the opinion that goats are not a threat to “nature” has gained consensus and appeared in scientific articles.]

Line 235: Are more recent (since 1980) estimates available?

Response: We have added a sentence to reflect the fact that “since then, the number of goats has not grown any further.” In line 272.

Line 237: Perhaps set off in quotes "everyday form of resistance"?

Response: We thank the reviewer for this reminder. We have added quotes and it is located now in lines 274.

Line 240: Persecution or prosecution?

Response: We thank the reviewer for this comment. We have changed the choice of words from “prosecution” to “persecution.”

Line 253: Put "racially" in quotes, or just superior?

Response: We thank the reviewer for this reminder. We have added quotes to “racially”. It is now in line 289.

Lines 280-282: For those relocated Palestinians who have employment, what are common occupations (and associated health hazards)?

Response: Thank you for this remark. We have added the following sentence “Palestinians workers are often concentrated in low paying sectors of the economy such as construction and industries (45) that carry high occupational health hazards such as falls, injuries and death (46). To lines 319 - 321.

Line 295: Can you clarify -- were the 40,000 slaughtered animals shown to be diseased or was this a "preventive" culling? Did the 40,000 compose 28% of the total, with the other 72% free of disease?

Response: Reference 9 states that “*Starting on 1993, a veterinary campaign aimed to eradicating the disease among small ruminants and consisting of a “test and slaughter” policy*

supplemented with administration of the B. melitensis Rev.1 strain vaccine to all ewe-lambs and female kid-goats was implemented countrywide. Throughout the campaign, more than 40,000 animals were slaughtered and 72% of the flocks were declared free of disease.”

In reference 2 of that paper the authors state that *“Throughout the campaign, in the interim programme 1635 flocks (59,901 ewes and goats) and in the eradication campaign 4,292 flocks (195,176 ewes and goats) were tested. More than 40,000 ewes and goats were slaughtered establishing around 72% of the flocks clear of the disease.”*

So, one might assume that the program had an interim part and a more extensive eradication campaign and most animals that were slaughtered were infected.”

Lines 301-311: Can you say what conditions made possible this successful, community-based collaborative effort in Taibe?

Response: As we state in line 342, the campaign was intersectoral and community based, as we wrote it involved local doctors, nurses, the veterinary service, and school officials. Here we believe lies the strength of such a program in creating trust with the community.

Reviewer #2 (Comments for the Author):

Thank you for submitting your article for publication. This is an important subject and a unique perspective. Provided below are some of the details that should be addressed in the manuscript.

Response: We thank the reviewer for these kind words.

Minor revisions

-there is no need to capitalize the B in brucellosis, except when referring to the scientific name, Brucella spp.

Response: We thank the reviewer for this comment. We have changed the capitalization throughout the text.

-line 32, Palestinian should be plural.

Response: We thank the reviewer for this remark. We have changed it to Palestinians.

-line 49 appears to be a smaller (or maybe not bolded?) font size compared to neighboring lines.

Response: We apologize for this. We have made the entire text in Times New Roman font, size 12.

-line 49, should reding be reading?

Response: We thank the reviewer for the sharp eye and this remark. We have changed it to “reading”.

-line 52, goats does not need to be capitalized

Response: We thank the reviewers for this comment. We have changed the capitalization.

-line 75, disease does not need to be capitalized

Response: We thank the reviewers for this comment. We have changed the capitalization.

-line 81, Brucella Melitensis should be Brucella melitensis

Response: We thank the reviewers for this comment. We have changed the capitalization to melitensis throughout the text.

-line 82, Brucella Abortus should be Brucella abortus

Response: We thank the reviewer for this comment and have changed the capitalization to abortus throughout the text.

-line 82, sheep does not need the "s" to be plural

Response: We apologize for this mistake and have changed it to sheep.

-line 84, is it 0.2/100 or 0.2/100,000? Same question for 33.5/100 or 33.5/100,000? Line 89 refers to the incidence out of 100,000, so my guess is that it should state 02. and 33.5 out of 100,000, but there is currently a decimal instead of a comma in 100,000.

Response: We thank the reviewer for this reminder. All incidences are per 100,000. We have edited that in lines 84 and 89.

-line 93, 100, 102 (and many other lines in paper): the font size appears to be smaller or not as bold in some of these sentences. Please check for consistency throughout the paper.

Response: We apologize for this and have made the entire text the same font and size.

-line 122, it should be brucellosis instead of Brucella

Response: We thank the reviewer for this comment and have changed it to brucellosis. It is now in line 154.

-line 125, B. melitensis should be italicized.

Response: We thank the reviewer for this comment and have italicized B.melitensis.

-line 130, could say "fresh goat milk" instead of freshly milked goat milk

Response: We thank the reviewer for this comment. We have changed the wording to fresh goat milk.

-lines 130-139, are there additional citations/references that can be used throughout this section, or is it all related to reference number 16?

Response: We thank the reviewer for this comment. The references for the section about Malta are those of 18-21 in the revised manuscript. Most of the arguments are repetitive throughout these papers and there are some more references, but we did not want to overload the paper with more references. We are happily willing to add more if the reviewers wish so.

-lines 136-139, please check verb tense and ensure consistency (line 138, doesn't should be don't/do not)

Response: We thank the reviewer for this comment. We have corrected the verb tense and changed “doesn’t” to “don’t”.

-line 173, could state "the first prime minister of Israel" or "Israel's first prime minister."

Response: Response: We thank the reviewer for this comment. We have changed the wording to “The first Israeli Prime Minister” in line 207.

-line 185, could state "their goats rapidly became the target. . ." instead of "but also their goats who rapidly became the target. . ."

Response: We thank the reviewer for this comment and have changed the sentence to “their goats rapidly became the target”.

-line 187, per cent should be percent; depending on the journal specifications, could also be written as 3%.

Response: We thank the reviewer for this comment and have changed it to percent.

-lines 192-203, are all details covered by references 32 and 33?

Response: Yes, these are two extensive papers on the subject including the Master’s thesis of one of the authors for Harvard Law School.

-line 198, goat should be plural

Response: We thank the reviewer for this comment and have changed it to “goats”.

-line 223, should state between 1948 and 1966 (or from 1948 to 1966) instead of "between 1948 to 1966."

Response: We thank the reviewer for this comment and have changed it to “between 1948 and 1966.

-line 228, remove "a" from end of proletarianization

Response: We thank the reviewer for this comment and the sharp eye. We have removed it.

-line 233, delete "put"

Response: We thank the reviewer for this comment. We have deleted it.

-line 233, do not need hyphen in goat-owners

Response: We thank the reviewer for this comment. We have deleted the hyphen

-lines 241-242, check word choice, tense and order

Response: We thank the reviewer for this remark and have changed the sentence to “goats and their owners were still not prioritized in public policies and had unfavorable accessibility to allocated grazing land or adequate veterinary supervision”.

-line 246, reduce should be reducing

Response: We thank the reviewer for this comment and have changed it to reducing.

**-line 247, sentence could read ". . .a Palestinian parliamentary member pushed the repeal. .
."**

Response: We thank the reviewer for this comment and have changed the sentence accordingly.

-line 260, remove "a" after "as"

Response: We thank the reviewer for this remark and have removed “a”.

-line 261, "is" should be "are"

Response: We thank the reviewer for this remark and have changed “is” to “are”.

-line 262, could just say "Cows" instead of The Cows

Response: We thank the reviewer for this remark. We have deleted “the” before cows.

-line 264, could add hyphen to cow based

Response: We thank the reviewer for this remark. We have added a hyphen accordingly.

-line 267, should say "became a cornerstone of. . ." or "one of the cornerstones" and could replace the second use of "became" in sentence with "as it developed into. . ."

Response: We thank the reviewer for this comment. We have changed the sentence to “This dairy industry became one of the cornerstones of the Israeli economy as it developed to be fully regulated, industrialized, and vaccinated.” In lines 302-304.

-line 268, Brucella abortus is the appropriate spelling.

Response: We thank the reviewer for this remark, and we have changed it to abortus.

-line 269, replace "has been" with "was"

Response: We thank the reviewer for this comment. We have replaced “has been” with “was”.

-line 275, add "s" to Palestinian

Response: We thank the reviewer for this comment. We have added “s” and it is now Palestinians.

-line 275, the authors toggle between "backyard" and "back yard." Select one and be consistent.

Response: We thank the reviewer for this remark. We have changed the wording to “backyard” throughout the text.

-line 277 and 281, authors toggle between socio-economic and socio economic. Could also use socioeconomic. Select one and be consistent.

Response: We thank the reviewer for this comment and have changed the wording to socioeconomic throughout the text.

-line 278, Israel should be Israeli

Response: We thank the reviewer for this comment. We meant to say that their homeland

became to be known as Israel from that point on. We have changed it to “became Israeli citizens”.

-line 280-282, please check word choice, tense and order

Response: We thank the reviewer for this comment. We have changed the sentence to “Until the present day, most Palestinians in Israel live in segregated towns that are at the bottom of the socioeconomic grading. These towns became areas of segregated and racialized poverty and unemployment.” in lines 316-318.

-line 283, use 283,000 instead of 283 thousand

Response: We thank the reviewer for this comment and we have changed it accordingly to 230,000.

-line 315, should it be "both the Palestinian and the goat"?

Response: We thank the reviewer for this comment, and we have changes it to “both the Palestinian and the goat”.

Major revisions

-Please ensure that font size and other formatting is consistent throughout the paper, as there are multiple places throughout the paper where the font size appears to change.

Response: We thank the reviewer for this comment and have took care of the font and size across paper.

-Please ensure that the spelling of Brucella spp. or brucellosis is used appropriately throughout the paper.

Response: We thank the reviewer for this comment and have ensured that the spelling is correct across the paper.

-Additional use of citations in certain parts of the paper may be warranted.

Response: We thank the reviewer for this comment. We have added several citations across the paper and are more than happy to add where the reviewer suggests.

February 21, 2022

Dr. osama Tanous
University of California, Davis
School of Public Health
One Shields Avenue.
Medical Sciences Bldg. 1C
Davis, CA 95616

Re: mSystems01499-21R1 (Organized Violence and Organized Abandonment Beyond the Human; The Case of Brucellosis among Palestinians in Israel)

Dear Dr. osama Tanous:

Thank you for submitting your manuscript to mSystems. We have completed our review and I am pleased to inform you that, in principle, we expect to accept it for publication in mSystems. However, acceptance will not be final until you have adequately addressed the reviewer comments.

The authors have done well to address the reviewer comments, and the reviewers and I agree on the quality of the revision. In principle, the reviewers and I agree that the manuscript is acceptable for publication, however, a number of minor corrections were suggested that would be helpful to include. Due to the number and type of suggestions, I felt that this would be easier to remedy in another revision rather than during proofing. Once the corrections are made and a revised manuscript is submitted, I anticipate accepting the manuscript without sending it out for another round of reviews.

Preparing Revision Guidelines

Sincerely,

Suzanne Ishaq

Editor, mSystems

Journals Department
Reviewer comments:

Reviewer #1 (Comments for the Author):

Thank you for your close attention and thoughtful responses to suggestions.

Reviewer #2 (Comments for the Author):

Thank you for making such thorough revisions to your paper. There are just a few minor details you may want to address. I have included these in the attached document.

Reviewer #3 (Comments for the Author):

The revised manuscript has been improved in terms of biological sense. However, this study could be further expanded the concept of ONE Health concept, which is related to geopolitical factors (10.3389/fvets.2018.00014) in the case of this study. There are many ways to control the infectious disease, for those unsettled ones, this study makes a good example for a new approach or understanding of HB in both Palestinians and probably later for Israelis, since they are in the same community. For that, the "One Health" concept should be introduced in this manuscript. Moreover, this manuscript should be polished within the context of biological sense, considering a professional microbiological journal, in particular for the control measurements (after Line 333-), there should be some practical solutions other than arguing, please add this information based on the ONE Health, concept.

Additional minor points:

Line 73-74. should be the place to insert the citation, regarding the foodborne transmission and zoonotic nature of Brucella.
10.3389/fvets.2020.00521

Line 85. "Palestinian citizens in Israel (PCIs)" this sentence itself contains "racialization". You should carefully define this concept since they are along with these places long before the establishment of Israel.

Line 102-104, there should be a reference here.

Line 154. should be colon?

Line 333---

Alternative Brucellosis control measurement could be added from a biological point of view.

references are below:

10.3389/fvets.2022.767198

10.3389/fvets.2021.763410

10.3389/fvets.2020.593683

Line 390. Many of these references should be reformatted. They are not in a corrected style.

Authors' Responses to Reviewers 1 and 2

Date: 20 January 2022

Manuscript Title: Organized Violence and Organized Abandonment Beyond the Human; The case of Brucella among Palestinians in Israel

The authors would like to sincerely thank the reviewers for their deep, thorough reading of the manuscript and their constructive feedback and suggestions. These thoughtful comments and suggestions have pushed us to extensively edit and sharpen our manuscript. We are resubmitting a version with highlighted changes. We have also edited the abstract and importance sections accordingly.

Kindly see below our point-by-point responses to the reviewers' comments.

Reviewer #1 (Comments for the Author):

Thank you for your close attention and thoughtful responses to suggestions.

Response: We thank the reviewer for the kind words again and the edits that have made our manuscript much clearer and sharper.

Reviewer #2 (Comments for the Author):

Thank you for making such thorough revisions to your paper. There are just a few minor details you may want to address:

Response: We thank the reviewer for the kind words and edits that have significantly improved our manuscript.

-Line 81, the genus and species should be italicized (*Brucella melitensis*)

Response: we thank the reviewer for this edit. We have italicized the genus and species accordingly.

-Line 82, the genus and species should be italicized (*Brucella abortus*)

Response: we thank the reviewer for this edit. We have italicized the genus and species accordingly.

-Line 148, “brucella” as the genus name should be capitalized and italicized (*Brucella*) or you could say brucellosis instead.

Response: we thank the reviewer for this edit. We have changed the word to brucellosis.

-Line 136, would recommend using full name/title: Dr. Suzanne Ishaq, Assistant Professor of Animal and Veterinary Sciences

Response: We thank the reviewer for this edit and we have added the full name and title accordingly.

-Line 159, *B. melitensis* should be italicized as the scientific name of the organism.

Response: we thank the reviewer for this edit. We have italicized the name of the organism accordingly.

-Line 286, “reduceing” should be reducing

Response: we apologize for the typo and thank the reviewer for this edit. We have corrected the spelling.

-Line 308, should be *Brucella abortus* (any scientific name should be italicized)

Response: we thank the reviewer for this edit. We have italicized the genus and species accordingly.

Reviewer #3 (Comments for the Author):

The revised manuscript has been improved in terms of biological sense. However, this study could be further expanded the concept of ONE Health concept, which is related to geopolitical factors (10.3389/fvets.2018.00014) in the case of this study. There are many ways to control the infectious disease, for those unsettled ones, this study makes a good example for a new approach or understanding of HB in both Palestinians and probably later for Israelis, since they are in the same community. For that, the "One Health" concept should be introduced in this manuscript. Moreover, this manuscript should be polished within the context of biological sense, considering a professional microbiological journal, in particular for the control measurements (after Line 333-), there should be some practical solutions other than arguing, please add this information based on the ONE Health, concept.

Response: We thank the reviewer for this suggestion and resource. We have introduced and critically discussed “One Health” concept from line 332 to 3577.

Additional minor points:

Line 73-74. should be the place to insert the citation, regarding the foodborne transmission and zoonotic nature of Brucella.

10.3389/fvets.2020.00521

Response: we thank the reviewer for this suggestion. We have added the mentioned reference to line 74.

Line 85. "Palestinian citizens in Israel (PCIs)" this sentence itself contains "racialization". You should carefully define this concept since they are along with these places long before the establishment of Israel.

Response: we thank the reviewer for this comment. It is detailed throughout the manuscript how the Zionist settler colonial project have transformed the Palestinians into a minority in their land, but we have added the following definition in lines 85 – 88 to clarify the matter:

“The Palestinian citizens in Israel (PCIs) are the native Palestinians that remained in their homeland after the majority of Palestinians were expelled in the 1948 Nakba, (catastrophe) and were not allowed return. They became a minority in their own homeland that came to be known as Israel.”

Line 102-104, there should be a reference here.

Response: there is reference number 13 in the line 103. The reference is:

Ghanem-Zoubi N, Eljay SP, Anis E, Paul M. Association between human brucellosis and adverse pregnancy outcome: a cross-sectional population-based study. *Eur J Clin Microbiol Infect Dis* 2018; 37:883–8. Available from <https://link.springer.com/article/10.1007/s10096-017-3181-7>

Line 154. should be colon?

Response: we thank the reviewer for this question. No, it is meant to be “colonial anxieties” referring to the anxieties of the British empire and troops from the diseases in the colonies. This was a common theme with colonial troops across colonies especially the “tropics” and India. There was a constant fear that the soldiers get sick and bring some diseases back to England. That’s why we called the section “local diseases and colonial anxieties”.

Line 333--- Alternative Brucellosis control measurement could be added from a biological point of view.

references are below:

10.3389/fvets.2022.767198

10.3389/fvets.2021.763410

10.3389/fvets.2020.593683

Response: We thank the reviewer for this valuable suggestion and resources. We have completely changed the last section of the manuscript, from line 332 to 414 and discussed these references critically. We think the article regarding the trans-border smuggling of animals and their product is fascinating. We have discussed it in the context of Israel – the occupied West Bank. If the journal guidelines allow, we believe that this photo conveys our message regarding smuggling of animals across highly militarized and surveilled borders. We believe it would be a good fit for the article.

<https://www.nytimes.com/2017/02/13/learning/whats-going-on-in-this-picture-feb-13-2017.html>

Line 390. Many of these references should be reformatted. They are not in a corrected style.

Response: we thank the reviewer for this revision. The references were inserted using the reference management software “Mendeley” and adjusted for mSystems style. The software makes some mistakes, and we can correct those in the proofreading process.

March 16, 2022

Dr. osama Tanous
University of California, Davis
School of Public Health
One Shields Avenue.
Medical Sciences Bldg. 1C
Davis, CA 95616

Re: mSystems01499-21R2 (Organized Violence and Organized Abandonment Beyond the Human; The Case of Brucellosis among Palestinians in Israel)

Dear Dr. osama Tanous:

Your manuscript has been accepted, and I am forwarding it to the ASM Journals Department for publication. For your reference, ASM Journals' address is given below. Before it can be scheduled for publication, your manuscript will be checked by the mSystems production staff to make sure that all elements meet the technical requirements for publication. They will contact you if anything needs to be revised before copyediting and production can begin. Otherwise, you will be notified when your proofs are ready to be viewed.

Publication Fees:

We recognize that the video files can become quite large, and so to avoid quality loss ASM suggests sending the video file via <https://www.wetransfer.com/>. When you have a final version of the video and the still ready to share, please send it to mSystems staff at mssystemsjournal@msubmit.net.

For mSystems research articles, if you would like to submit an image for consideration as the Featured Image for an issue, please contact mSystems staff at mssystemsjournal@msubmit.net.

Sincerely,

Suzanne Ishaq
Editor, mSystems

Journals Department
Phone: (202) 737-3600